# Methodologies for studying depression in persons living with tuberculosis: Protocol for a scoping review

Amanda J. Gupta[1,2]*, Patricia Turimumahoro[3], Lori Rosman[4], Jonathan E. Golub[5,6,7], David W. Dowdy[5,6]

**1** Epidemiology of Microbial Diseases, Yale School of Public Health, New Haven, Connecticut, United States of America, **2** Johns Hopkins Bloomberg School of Public Health, Maryland, United States of America, **3** World Alliance for Lung and Intensive Care Medicine in Uganda (WALIMU), Kampala, Uganda, **4** Welch Medical Library, School of Medicine, Johns Hopkins University, Baltimore, Maryland, United States of America, **5** Department of Epidemiology, Johns Hopkins Bloomberg School of Public Health, Baltimore, Maryland, United States of America, **6** Department of International Health, Johns Hopkins Bloomberg School of Public Health, Baltimore, Maryland, United States of America, **7** Center for Tuberculosis Research, Johns Hopkins University School of Medicine, Baltimore, Maryland, United States of America

* amanda.meyer@yale.edu

## Abstract

Tuberculosis (TB) and depression frequently co-occur, yet research has largely focused on prevalence rather than diagnostic or treatment methodologies. Given overlapping symptoms, robust research approaches are critical for improving detection and treatment strategies in persons with TB (PWTB). This protocol defines a scoping review that aims to map methodologies used to study depression in individuals with TB, identifying gaps in research design, diagnosis, and treatment that may hinder clinical and public health advancements. A search was conducted in MEDLINE, Embase, PsycINFO, Global Health, Cochrane Library, and Africa-Wide Information using controlled vocabulary related to TB and depression. Studies examining the TB-depression relationship will be included. We will use Covidence to facilitate screening, selection, and data extraction. We will extract data on study design, diagnostic tools, treatment interventions, and analytical approaches. Descriptive characteristics of included studies will be presented using figures and tables. Using the Reach-Effectiveness-Adoption-Implementation-Maintenance (RE-AIM) framework, we will evaluate reach by assessing the extent to which studies explore both directions of the TB-depression relationship and include diverse, high-risk populations. Effectiveness will be examined by categorizing study designs to assess methodological diversity and evaluating the diagnostic tools and treatment interventions used, along with their reported efficacy. Adoption will be analyzed by identifying where and by whom depression diagnosis and treatment methodologies have been implemented and whether research findings have influenced clinical guidelines or public health policies. Implementation will be assessed by identifying barriers and facilitators reported in qualitative studies or by researchers

**Data availability statement:** No datasets were generated or analysed during the current study.

**Funding:** The author(s) received no specific funding for this work.

**Competing interests:** The authors have declared that no competing interests exist.

regarding the integration of depression care into TB treatment settings. Finally, maintenance will be determined by examining whether diagnostic and treatment methodologies were sustained within health systems, including long-term patient outcomes and the persistence of interventions beyond the research phase. By mapping existing methodologies and identifying research gaps, this review will provide valuable insights to guide future research study designs and improve diagnostic and treatment strategies for depression in PWTB.

## Introduction

Tuberculosis (TB), a communicable disease caused by *Mycobacterium tuberculosis,* is one of the leading infectious causes of mortality worldwide [1]. While it primarily affects the lungs (pulmonary TB), it can be found in other places throughout the body (extrapulmonary TB). The COVID-19 pandemic accelerated deaths due to TB as less attention, and subsequently funding, has been paid to TB [1,2]. This resulted in ~10.8 million people developing new cases of active TB in 2023 with ~1.25 million deaths due to TB disease [1]. Active TB refers to disease in which TB bacteria are replicating and causing clinical symptoms. Pulmonary TB is typically symptomatic and infectious while extrapulmonary TB is usually symptomatic but not transmissible. This contrasts with latent TB, a state in which an individual is infected but asymptomatic and non-infectious. It is estimated that billions of individuals worldwide have latent TB [1]. Furthermore, TB can be classified by drug resistance with ~400,000 persons with TB (PWTB) having multidrug-resistant TB (MDR-TB) in 2023 [1]. Despite this large burden, TB is vastly underfunded leading to under-supported research endeavors and national TB programs left without appropriate funding to successfully combat the large TB burdens that they may face [1–5]. This results in gaps in TB diagnosis, treatment, and care in high TB burden settings [6–8].

One such gap in care is in the screening, diagnosis, and treatment of comorbidities in PWTB [9]. While the diagnosis and treatment protocols of infectious comorbidities of TB such as Human Immunodeficiency Virus (HIV) have been extensively researched [10–13], the diagnosis and treatment of comorbid mental health disorders have received little attention, despite their impacts on TB treatment [9,14,15]. Depression is one such overlooked comorbidity with potentially large impacts on PWTB [16,17]. Depression, a common mental health condition characterized by persistent sadness, loss of interest or pleasure, and functional impairment, can affect quality of life if not properly managed [18]. While clinically defined in various tools, its expression and recognition vary across cultural contexts, increasing its diagnostic and treatment complexity [19].

Depression appears to impact PWTB more frequently than the general population. One systematic review estimates the depression prevalence amongst PWTB to be 45% [20]. Depression may not only be a consequence of TB, but also a risk factor due to its effects on immune function, health-seeking behavior, and treatment engagement [15,21,22]. TB-depression co-occurrence can be thought of as a

syndemic, or an interacting set of conditions that cluster within social disadvantage (e.g., stigmatization of health conditions and economic vulnerability) that results in an increased burden of disease [23–25]. The burden of depression may also vary by TB subtype. For example, individuals with extrapulmonary TB may face diagnostic uncertainty that heightens psychological distress [26,27]. Those with MDR-TB often endure prolonged treatment, more severe side effects, greater economic uncertainty, and greater social isolation, all of which may increase the risk or severity of depression [28,29].

Given this complex interplay, most PWTB are never screened for depression, let alone receive treatment for depression. This is due to a lack of providers, funding, cultural norms, and stigma [20,22,30,31]. From a public health perspective, untreated depression contributes to delays in TB diagnosis and treatment initiation [30,32], non-adherence to TB medications [33,34], and ultimately increases negative TB outcomes such as loss to follow up and death [35,36].

Despite the significant impact of depression in PWTB, research has largely been limited to prevalence surveys. Few studies have used longitudinal designs and even fewer explore the methodologies used to screen, diagnose, or treat depression in PWTB. This may be in part due to diagnostic challenges posed by overlapping symptoms, such as TB-related fatigue, weight loss, and general malaise, all of which can masquerade as signs of depression, leading to misdiagnoses or missed diagnoses.

Given the substantial burden of depression among PWTB and the large focus on prevalence estimates within the literature, a scoping review is particularly needed to broadly map the existing evidence, identify the most pressing knowledge gaps, and clarify how research has been conducted when studying these comorbid conditions. Therefore, we aim to conduct a scoping review to examine the methodologies used to study depression in PWTB, including diagnostic and treatment methodologies, identifying research gaps and highlighting opportunities for improved methodological approaches.

## Methods

This section is presented in accordance with the scoping review framework by Arskey and O'Malley [37] and refined by Levac et al [38] which guided the methodology of our scoping review. This framework consists of six steps: 1) Identifying the research question 2) Identifying relevant studies 3) Study selection 4) Charting the data 5) Collating, summarizing, and reporting the results and 6) Consultation with stakeholders on study findings.

### Identifying the research question

The research question for our review is: what methodologies have been used in the research literature to study depression (including diagnosis and treatment) among individuals with tuberculosis, and what are the gaps in methodological approaches?

### Identification of relevant studies

Table 1 includes the eligibility criteria for this scoping review. We searched the following electronic databases: MEDLINE, Embase, Global Health, the Cochrane Library, the WHO Regional Libraries, Africa-Wide Information, and PsycINFO. These databases were selected for their relevance to the scope of this review: MEDLINE, Embase, Global Health, and the Cochrane Library are widely used for biomedical and public health research; the WHO Regional Libraries and Africa-Wide Information are included to ensure geographic representation, particularly from high TB burden regions such as sub-Saharan Africa; and PsycINFO is included for its comprehensive coverage of psychological and mental health research. The preliminary search strategy, constructed by a trained medical librarian (LR) after several weeks of testing various combinations of terms to ensure maximum reach, focused on combinations of terms related to "tuberculosis" and terms related to "depression" (S1 File).

We did not include date restrictions. The electronic database search will be supplemented by manual searches, including the screening of reference lists from all included studies to identify any additional relevant articles that may have been initially missed.

**Table 1. Study eligibility criteria.**

| Domain | Included | Excluded |
|---|---|---|
| Participants | PWTB regardless of age, sex, socioeconomic status and TB type including:<br>• Microbiologically confirmed<br>• Clinically diagnosed<br>• Pulmonary<br>• Extra-pulmonary<br>• Latent<br>• Active<br>• Drug-resistant<br>• Drug-sensitive | • Those without a confirmed tuberculosis diagnosis<br>• Animal participants<br>• Participants with nontuberculous mycobacterial infections (e.g., leprosy)<br>• Studies that only include laboratory samples and isolates |
| Concept or intervention | Studies that contain information on depression and tuberculosis including:<br>• Diagnostic tools, methods, and protocols<br>• Therapeutic interventions (pharmacological or psychological)<br>• Determinants of the TB-depression syndemic and underlying pathways and mechanisms | • Studies that explore tuberculosis or depression separately<br>• Studies that only focus on biological mechanisms of tuberculosis and/or depression |
| Outcome | • Sensitivity, specificity, and/or accuracy of depression diagnostic tools in PWTB<br>• Studies focused on the identification of diagnostic challenges, barriers, or facilitators<br>• Validation of depression diagnostic measures in PWTB<br>• Changes in depression symptoms (including patient-reported outcomes) during TB treatment<br>• TB treatment adherence, completion, and/or complication rates because of depression treatment or lack thereof<br>• Studies with outcomes on methodological challenges in studying depression and tuberculosis concurrently | Outcomes not explicitly linked to the study of depression in PWTB including:<br>• Studies reporting biochemical, genetic, or molecular outcomes without clinical or diagnostic relevance<br>• Studies with general TB treatment outcomes unrelated to or not considering depression<br>• Studies with insufficient detail on how outcomes were measured or analyzed<br>• Studies based solely on opinions or informal observations |
| Context | Research conducted in healthcare or community settings of relevance such as<br>• Tuberculosis clinics<br>• Hospital settings<br>• Primary care clinics<br>• Community health programs<br>• Any other relevant setting where tuberculosis and depression services may be provided<br>Global geographic inclusion | Research from unrelated contexts such as veterinary or laboratory studies.<br>No exclusion criteria based on geography |
| Study Design | • Primary studies<br>• Randomized controlled trials<br>• Prevalence surveys<br>• Case studies<br>• Cohort studies<br>• Case-control studies<br>• Systematic reviews<br>• Meta-analyses<br>• Scoping reviews<br>• Qualitative studies<br>• Mixed methods analyses<br>• Diagnostic test accuracy studies | • Meta-reviews (reviews of reviews) and umbrella reviews<br>• Opinion pieces, editorials, and letters to the editor that do not contain empirical data<br>• Abstracts without full-text availability<br>• Protocol papers |
| Publication Language | English | Language other than English |

## Study selection

We are using Covidence to screen study titles, abstracts and full text articles for inclusion. Two reviewers (AJG, PT) will review abstracts in parallel for inclusion. To minimize bias, reviewers will be blinded to each other's ratings except when resolving conflicts. Reviewers will meet after every 20 abstracts to resolve any conflicts and to propose/implement

updates to the search strategy or inclusion criteria. All abstracts that pass screening will have the full text reviewed by the two reviewers, again, in parallel for final inclusion in the review. Disagreements will be resolved via discussion and, if needed, by a third reviewer (DWD).

### Charting the data

After piloting the data extraction tool, we will extract data using Covidence. We will extract pertinent study details which can be found in Table 2. Additionally, if the authors identify any limitations relevant to the research question, such as methodological limitations, we will extract this data. One author (AJG) will extract the data with a second reviewer (PT) checking extracted data for accuracy. These two authors will routinely meet during the data extraction phase to ensure the extraction tool is capturing all relevant data needed to answer the overall research question.

## Results

### Collating, summarizing, and reporting results

We will provide descriptive characteristics of all included studies using figures and tables to fully summarize the studies, stratifying where appropriate by key variables such as by age, TB type, geographic location and/or study design to name a

**Table 2. Data extraction.**

| Category | Data Items to Extract |
|---|---|
| Study Identification | • Author(s)<br>• Year of Publication<br>• Journal Name |
| Study Context | • Geography<br>• Setting (e.g., community, health facility)<br>• Timeframe of Study |
| Study Design & Objectives | • Research Aim<br>• Study Design (e.g., cross-sectional, RCT, qualitative) |
| Population Characteristics | • Sample Size<br>• Age<br>• Sex<br>• HIV Status<br>• Socioeconomic Status<br>• Other Mental Health Conditions |
| Depression Screening/Diagnosis | • Screening/Diagnostic Tools Used (e.g., PHQ-9, clinical interview) |
| Intervention Characteristics | Type:<br>• Psychological (e.g., CBT)<br>• Pharmacological (e.g., antidepressants)<br>• Alternative (e.g., mindfulness) |
| | Components:<br>• Mode of Delivery (e.g., in-person, oral)<br>• Duration<br>• Frequency |
| Analytical Approach | • Statistical Methods (e.g., regression, chi-square)<br>• Comparison Population (if applicable) |
| Outcomes Assessed | • Outcome Measures (e.g., prevalence, odds ratios)<br>• Measures of Precision (e.g., 95% confidence intervals)<br>• Statistical Significance |
| Qualitative Study Data | • Key Themes Identified<br>• Barriers/Facilitators to Diagnosis or Treatment<br>• Analytical Framework (e.g., thematic analysis, grounded theory, case study) |

few. While we will not consider literature quality among inclusion criteria, we will conduct a quality appraisal for each study as the presence or absence of high-quality research is relevant to understanding research gaps. As there will be many different study designs, to assess quality, we will use the Jadad scale for randomized controlled trials, the Newcastle-Ottawa scale for observational studies, and the Joanna Briggs Institute Critical Appraisal Tool for Qualitative Research.

We will structure the reporting of our results using the RE-AIM (Reach, Effectiveness, Adoption, Implementation and Maintenance) framework [39]. For _reach_, we will determine the extent to which studies have explored both directions of the TB-depression relationship. We will also determine the _reach_ of researchers in terms of including representative samples of PWTB and their inclusion of high-risk groups (such as those with HIV, drug-resistant TB, or pediatric cases). For _effectiveness_ we will categorize the study designs used to assess the diversity of methodological approaches. A wide range of designs – such as cross-sectional, longitudinal, and randomized– is essential for fully elucidating the co-occurrence of TB and depression. A lack of diverse study designs may indicate gaps in the field's ability to fully understand this relationship. Additionally, we will determine the types of diagnostic measures used for depression along with their effectiveness (if reported) as well as how efficacious or effective any treatment interventions were at impacting depression- or TB-outcomes.

For _adoption_ we will look at the extent to which depression screening, diagnosis, and treatment methodologies are being adopted in different settings (such as by national programs, referral centers, local clinics, etc) and by whom (e.g., physicians, nurses, community health works etc.) and examine the way in which research findings on these two diseases have impacted clinical guidelines and public health policies if mentioned by study authors. For _implementation_, we will determine barriers and facilitators identified to the implementation of depression related care in real-world TB treatment settings. Finally, for _maintenance_ we will determine whether any depression related methodologies for diagnosis or treatment were sustained or embedded within existing health systems. This will include, if reported upon, long-term patient outcomes, if methodologies persisted beyond the research phase, and if any studies evaluated how health systems integrated and sustained any proposed diagnostic or treatment interventions.

## Consultation with stakeholders

We will include a consultation phase to enhance the relevance and applicability of our findings. We will engage a diverse group of stakeholders, drawing on global collaborators with expertise in TB and mental health from an established network of the research team, as well as individuals with lived experience of TB, including PWTB from Uganda who are participating in an unrelated, ongoing cohort study. Consultations will offer contextual insights and help identify gaps or implementation issues not captured in the published literature, ensuring the findings are meaningful and actionable for high TB burden settings.

## Data sharing

In line with open science principles, we will make the final data extraction sheet and study metadata openly available through a public repository such as the Open Science Framework upon completion of the review.

## Timeline

The initial search was conducted on March 31, 2025. All screening, including title and abstract screening and full text screening are taking place from April to June 2025. Data extraction will take place in July 2025. Analysis and synthesis of results will take place from July to August 2025 with final results expected by September 2025.

## Discussion

This scoping review will explore the existing literature on the intersection between depression and TB allowing us to identify research gaps. Importantly, this will allow us to understand methodological gaps that exist and approaches that researchers have historically taken to tackle these two diseases concurrently.

Findings from this review could contribute to advancing research beyond cross-sectional prevalence surveys by encouraging the use of other quantitative, qualitative, and mixed-methods study designs that can better inform evidence-based practice. Furthermore, this review could underscore the urgent need for validated diagnostic tools and specialized treatment approaches to address overlapping symptoms between TB and depression, ultimately supporting the development of more effective strategies.

Strengths of our review will include methodological rigor as we are using the Levac et al framework to ensure a systematic and transparent approach to study identification and selection. This will be enhanced through the use of RE-AIM to synthesize results, providing a structured lens to assess how methodological approaches to researching TB and depression have created a knowledge base as well as gaps in studying the diseases concurrently. Finally, the planned stakeholder consultations will add depth and practical relevance to the final interpretation of findings, ensuring methodological rigor and relevance to global TB and mental health research practices.

Our review will have some limitations that are worth discussing a priori. First, we have limited the search to English language studies. Given the burden of tuberculosis in non-English speaking countries, this may exclude several studies that would otherwise be eligible. Second, we are beginning this search with the term depression, a Western term, and its associated synonyms. Many countries use alternate terms for depression, such as shenjing shuairuo (nervous weakness) in China or yuutsu in Japan, meaning that our initial search may not fully capture the available literature. Given the long list of terms that could be used by different countries, it is not likely we will capture all possible terms and subsequently may miss eligible publications. Third, scoping reviews cannot make definitive conclusions about intervention effectiveness or causality given their focus on describing existing literature. However, it may be appropriate to conduct a subsequent systematic review and meta-analysis to assess public health impact of depression-related interventions amongst PWTB. Finally, as with all scoping reviews, the breadth of study types and topics can make data synthesis more complex than in a systematic review. We believe our use of RE-AIM to structure the results, regardless of study type, will aid in creating meaningful outputs.

## Conclusions

This scoping review will advance understanding of how depression screening, diagnosis, and treatment have been studied in the context of tuberculosis by systematically mapping existing methodologies and identifying critical research gaps to guide future public health research and intervention development.

## Supporting information

**S1 File: Initial Search terms by database.**
(DOCX)

**S2 File: PRISMA-P Checklist.**
(DOCX)

## Acknowledgments

We would like to thank both Yale University and Johns Hopkins University for providing institutional support for this work.

## Author contributions

**Conceptualization:** Amanda J. Gupta.

**Data curation:** Patricia Turimumahoro.

**Formal analysis:** Amanda J. Gupta, Patricia Turimumahoro.

**Methodology:** Amanda J. Gupta, Lori Rosman, Jonathan E. Golub, David W. Dowdy.

**Project administration:** Amanda .J Gupta, Patricia Turimumahoro, Lori Rosman, David W. Dowdy.

**Writing – original draft:** Amanda J. Gupta.

**Writing – review & editing:** Amanda J. Gupta, Patricia Turimumahoro, Lori Rosman, Jonathan E. Golub, David W. Dowdy.

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
