## [Decision Letter · Decision Letter 0]

PONE-D-25-16622Methodologies for Studying Depression in Persons living with Tuberculosis: Protocol for a Scoping ReviewPLOS ONE

Dear Dr. Gupta,

Thank you for submitting your manuscript to PLOS ONE. After careful consideration, we feel that it has merit but does not fully meet PLOS ONE’s publication criteria as it currently stands. Therefore, we invite you to submit a revised version of the manuscript that addresses the points raised during the review process.

We look forward to receiving your revised manuscript.

Kind regards,

Mickael Essouma, M. D.

Academic Editor

PLOS ONE

Journal Requirements:

**Additional Editor Comments:**

My comments are appended in the attached PONE-D-25-16662_Mickael Essouma document.

Mickael Essouma

Reviewers' comments:

Reviewer's Responses to Questions

**Comments to the Author**

1. Does the manuscript provide a valid rationale for the proposed study, with clearly identified and justified research questions?

Reviewer #1: Yes

Reviewer #2: Yes

2. Is the protocol technically sound and planned in a manner that will lead to a meaningful outcome and allow testing the stated hypotheses?

Reviewer #1: Yes

Reviewer #2: Partly

3. Is the methodology feasible and described in sufficient detail to allow the work to be replicable?

Reviewer #1: Yes

Reviewer #2: Yes

4. Have the authors described where all data underlying the findings will be made available when the study is complete?

Reviewer #1: Yes

Reviewer #2: Yes

5. Is the manuscript presented in an intelligible fashion and written in standard English?

Reviewer #1: Yes

Reviewer #2: Yes

6. Review Comments to the Author

You may also provide optional suggestions and comments to authors that they might find helpful in planning their study.

Reviewer #1: Review for: “Methodologies for Studying Depression in Persons living with Tuberculosis: Protocol

for a Scoping Review“

The paper at hand is the study protocol for a scoping review about methodologies for studying depression in persons suffering from tuberculosis. The paper is generally well written. The authors refer to established guidelines of conducting scoping reviews. The inclusion of supporting information like the PRISMA checklist or the search terms is laudable. The high prevalence of depression in tuberculosis and the detrimental consequences of both conditions makes this a very important scientific endeavor. In my view this work shows great promise and I applaud the authors for their work. Nevertheless, I have some concerns and comments that I feel need to be addressed. Please find my detailed review below.

My main point concerns the description of the aim of the planned scoping review. The methods section mostly seem to indicate that the aim is to find treatment trials, i.e., trials in which either depression and/or tuberculosis are treated. In the results section you convey that you will consider diverse study types. This seems good because treatment studies are not the only way to find out more about any associations between the two diseases. In any case, after the revision, the paper should give some examples for typical types of studies that you aim to include. I get that you cannot anticipate every type of study that explores the interrelationships between tuberculosis and depression but at the moment it is hard to grasp how do you aim to get relevant information on comorbidity and treatment options and whether interventional studies are the backbone of your review or one source of many. I don't think there is necessarily a problem with the content here, but rather difficulties arise due to the manner in which the research plan and the narrative flow of the text are presented.

The second major point that strikes me is the theoretical embedding of the connections between tuberculosis and depression. It is quite well deduced that it exists and that it should be investigated further. However, it does not go into what previous findings have shown as to why this connection exists. I am not an expert in this field. However, I wonder what psychological and/or biological factors could be interacting here? Is there relevant biopsychological literature regarding the interrelationship? What about the neuroscience of tuberculosis?

I also wonder whether meta-theories such as the “complex systems” perspective (which I am more familiar with) could be utilized to explain the relationships?

Fried, E. I., & Robinaugh, D. J. (2020). Systems all the way down: embracing complexity in mental health research. BMC medicine, 18, 1-4.

Scheffer, M., Bockting, C. L., Borsboom, D., Cools, R., Delecroix, C., Hartmann, J. A., ... & Nelson, B. (2024). A dynamical systems view of psychiatric disorders—theory: a review. JAMA psychiatry, 81(6), 618-623.

Westhoff, M., Berg, M., Reif, A., Rief, W., & Hofmann, S. G. (2024). Major problems in clinical psychological science and how to address them. Introducing a multimodal dynamical network approach. Cognitive Therapy and Research, 48(5), 791-807.

More importantly, there seem to be reviews that discuss possible reasons for mental health problems in tuberculosis patients that you could cite (or discuss in more depth).

Cite:

Alene, K. A., Wangdi, K., Colquhoun, S., Chani, K., Islam, T., Rahevar, K., ... & Viney, K. (2021). Tuberculosis related disability: a systematic review and meta-analysis. BMC medicine, 19, 1-19.

Discuss in more depth:

Hayward, S. E., Deal, A., Rustage, K., Nellums, L. B., Sweetland, A. C., Boccia, D., ... & Friedland, J. S. (2022). The relationship between mental health and risk of active tuberculosis: a systematic review. BMJ open, 12(1), e048945.

Cáceres, G., Calderon, R., & Ugarte-Gil, C. (2022). Tuberculosis and comorbidities: treatment challenges in patients with comorbid diabetes mellitus and depression. Therapeutic Advances in Infectious Disease, 9, 20499361221095831.

I get that an extensive discussion might be to much for a protocol paper but some hints into the possible nature of the relationships should be given and be it on a meta (e.g., systems) level. For a protocol paper, I think that one to two paragraphs are enough but the reader will need some context information here.

Finally, the protocol should have a statement about open science practices. Will the extraction sheet be made openly available? How do you plan to share important study metadata? If you aim to share your data, at which repository do you want to share it? If you do not aim to share your data, this deserves justification in my view.

Minor

Sometimes you seem to use for indicating your sources and sometimes you use (). Please stick to a standardized style of reference indicators.

The list of references is quite short at the moment. I am aware that this is a protocol paper but I would think that the paper would benefit from additional sources.

Line 55: “is one of the leading infectious killers worldwide”: I would recommend against using “killer” and instead use words like lethality / mortality here. Furthermore, I would recommend you cite relevant literature to support this (true) claim.

Line 74: Please remove the “staggering”.

Table 1 contains long sentences and is really difficult to read. I think it needs substantial rework. I would recommend that you shorten the text, consider using bullet points instead of sentences and consider using spacing options to make your points clearer.

Line 115: Good that you screen the references of the included papers. Consider including relevant literature for the benefits

of footnote chasing.

Limitations section: I would recommend that you critically discuss the strength and shortcomings of scoping reviews compared to traditional systematic reviews and meta-analyses and justify your choice for the scoping review. What I like about the limitations section is the critical discussion of cultural aspects.

Line 219: “Our review does have”: Maybe write: “Our review will have some limitations that are worth pointing out a priori”... Right now it sounds as if the review was already conducted. Down below in the conclusions section you use “will” and I would recommend also using it here.

I support an open and transparent review process. Therefore, I do not wish to stay anonymous.

Thanks to the authors and to the journal editors for giving me the opportunity to review this paper.

Max Berg,

University of Marburg

Clinical Psychology Group

Reviewer #2: The article is a study protocol for a scoping review aimed at mapping methodologies used to study depression in individuals with tuberculosis (TB). The primary objective is to identify gaps in research design, diagnostic tools, and treatment strategies for depression in TB patients. The authors highlight the co-occurrence of TB and depression and the need for robust research approaches to improve integrated care. The protocol outlines a systematic approach to reviewing existing literature, with the goal of guiding future research and public health interventions.

Methodology Evaluation:

The study employs the Arksey and O’Malley framework, updated by Levac et al., for conducting scoping reviews, which is appropriate for mapping broad research areas. The protocol includes six steps: identifying the research question, selecting relevant studies, charting data, collating results, and consulting stakeholders. The search strategy is comprehensive, covering databases like MEDLINE, Embase, PsycINFO, and Africa-Wide Information, ensuring a global perspective. Inclusion criteria are well-defined, focusing on studies that examine the TB-depression relationship, while exclusion criteria eliminate irrelevant or low-quality studies.

Data extraction will be performed using Covidence, with two reviewers screening abstracts and full texts to minimize bias. The RE-AIM framework (Reach, Effectiveness, Adoption, Implementation, Maintenance) will structure the analysis, which is a strength for evaluating the impact of research methodologies. However, the protocol limits the search to English-language studies, potentially excluding relevant research from high-TB-burden, non-English-speaking countries. Additionally, the reliance on Western terms like "depression" may overlook culturally specific expressions of mental health conditions.

Strengths and Weaknesses:

Strengths:

Comprehensive search strategy across multiple databases.

Clear inclusion/exclusion criteria to ensure relevance.

Use of the RE-AIM framework to evaluate methodological impact.

Dual-reviewer process to reduce bias in study selection.

Weaknesses:

Language restriction (English-only) may exclude significant studies.

Potential cultural bias in defining depression, missing non-Western perspectives.

No pilot data or preliminary results to validate the approach.

Limited discussion of how heterogeneity in study designs will be managed during synthesis.

Conclusion and Recommendations:

This protocol provides a rigorous framework for a scoping review that could significantly advance understanding of depression in TB patients. However, the exclusion of non-English studies and culturally specific terminology may limit its comprehensiveness. Future research should:

Expand language inclusion to capture studies from high-TB-burden regions.

Incorporate culturally adapted definitions of depression.

Pilot the methodology to refine search terms and data extraction processes.

Address heterogeneity in study designs during data synthesis.

Overall, the protocol is well-structured and promises valuable insights, but its impact could be enhanced by addressing these limitations.

7. PLOS authors have the option to publish the peer review history of their article (what does this mean? ). If published, this will include your full peer review and any attached files.

**Do you want your identity to be public for this peer review?** For information about this choice, including consent withdrawal, please see our Privacy Policy .

Reviewer #1: **Yes: ** Max Berg

Reviewer #2: **Yes: ** ABENA FOE Jean-Louis

---

## [Author Response · Author response to Decision Letter 1]

16 May 2025

Dear Reviewers

We thank the review team for their careful consideration of our protocol manuscript. We believe the reviewers suggestions will improve both this protocol manuscript as well as our scoping review. Below we have responded to the reviewers suggestions as well as made changes where appropriate to the manuscript itself.

Best

Amanda J Gupta

Journal Requirements:

Please note, we have updated the formatting to meet PLOS ONE’s style requirements. Thank you for directing us to the appropriate guides for doing so.

We have updated the data availability statement to indicate that in the present submission, there is no data generated as this is a protocol manuscript.

Additional Editor Comments:

Consider complying with PLOS One guidelines for formatting manuscripts. Accordingly, the authors are urged to not subdivide the abstract into sub-sections. Furthermore, the authors are not expected to present results in any section of the scoping review protocol article given that the study results are not expected at this stage of the research process.

Thank you. We have updated the abstract to reflect this guidance. While this study itself has no results, we still believe there is a benefit in having a results section for the main body of the manuscript as it outlines how we will report the results of our scoping review.

I appreciate the figures provided regarding the epidemiologic and economic burden of TB. However, the authors could strengthen this paragraph by starting the introduction with the definition and classification (by the type of organ involvement: pulmonary versus extrapulmonary [specification of the extrapulmonary body organs that can be afflicted by TB required] TB;by disease activity: active TB versus latent TB; and by drug resistance: MDR-susceptible TB and MDR-resistant TB of tuberculosis) of TB, especially since PLOS One is a generalist journal. This would set the stage for the authors to make comments about TB throughout the manuscript (not just in the introduction) based on its subtypes, because there are indeed differences between pulmonary TB and extrapulmonary TB in the one hand, and between MDR-susceptible TB and MDR-resistant TB on the other hand. As a result, there may also bedifferences in the burden of depression across TB subtypes

Thank you for making this point. To ensure all your readers, including those not as well versed in TB, understand the burden that TB poses as well as the disease itself, we have updated the intro paragraph (lines 65-82) to the following to highlight the different types of TB along with their burden:

Tuberculosis (TB), a communicable disease caused by Mycobacterium tuberculosis, is one of the leading infectious killers worldwide. While it primarily affects the lungs (pulmonary TB), it can be found in other places throughout the body (extrapulmonary TB) such as in the lymph nodes, pleura, bone and joints, urogenital tract, and meninges to name a few. The COVID-19 pandemic accelerated deaths due to TB as less attention, and subsequently funding, has been paid to TB (1, 2). This resulted in an estimated 10.8 million people developing new cases of active TB in 2023 with an estimated 1.25 million deaths due to TB disease (2). Active TB refers to disease in which TB bacteria are replicating and causing clinical symptoms. Pulmonary forms of active TB are typically symptomatic and infectious while extrapulmonary TB is usually symptomatic but not transmissible. This contrasts with latent TB which is a state in which an individual is infected with TB but is generally asymptomatic and non-infectious. It is estimated that billions of individuals worldwide have latent TB. Furthermore, TB can be classified by drug resistance with ~400,000 PWTB having multidrug-resistant TB (resistance to at least isoniazid and rifampin) in 2023 (2). Despite this large burden, TB is vastly underfunded leading to under-supported research endeavors and national TB programs left without appropriate funding to successfully combat the large TB burdens that they may face (1-5). This results in gaps in TB diagnosis, treatment, and care in high TB burden settings (6-8).

In addition, we added in the following sentence (line 111-126) to reflect the fact that there may exist differences in depression burden by TB subtype:

Additionally, the burden of depression may also vary by TB subtype —individuals with extrapulmonary TB may face diagnostic delays and uncertainty that heighten psychological distress (26,27), while those with multidrug-resistant TB (MDR-TB) often endure prolonged treatment, more severe side effects, and greater social isolation, all of which may increase the risk or severity of depression (28,29).

Consider clarifying this statement. The revised statement could read: "While the diagnosis and treatment protocols of infectious TB comorbidities such as Human Immunodeficiency Virus (HIV) infection have been extensively researched, the diagnosis and treatment of comorbid non-communicable diseases (NCDs) such as mental health disorders have received little attention, even as the identification and treatment of comorbid NCDs has been shown to positively impact the TB outcome in affected patients."

We have updated the sentence to incorporate your suggested wording (see lines 85-92).

This sentence needs to be supported by at least one reference. Furthermore, it would be great to provide a multiperspective definition of depression after this sentence.

Suggested references:

1. PMID: 38357431 PMCID: PMC10863678 DOI: 10.32872/cpe.11699

2. https://doi.org/10.1016/j.socscimed.2016.12.030

3. PMID: 38856993 DOI: 10.1001/jama.2024.5756

Thank you for not only the suggestion but providing wonderful citations to go along with it! We have added in two TB specific citations for the original sentence in question (line 95) and have now added the following sentences (lines 95-100):

Depression is a common mental health condition characterized by persistent sadness, loss of interest or pleasure, and functional impairment which can affect the overall quality of life of individuals suffering from it if not properly managed (18). While clinically defined in various tools, such as the DSM-5, its expression and recognition vary across cultural and social contexts increasing the diagnostic and treatment complexity of the condition (19).

Consider revising this paragraph to address the mechanisms (including the high frequency of HIV infection in the population with TB) underpinning the TB-depression syndemic, challenges in and facilitators for diagnosing and treating depression across TB subtypes, and the public health impact of diagnostic and therapeutic interventions targeting depression in individuals with TB.

In addition to the references provided by reviewer 1, these references could be helpful for this purpose:

1. PMID: 34140898 PMCID: PMC8203803 DOI: 10.3389/fpsyt.2021.617751

2. Wang J, Wu X,

Lai W, et al. Prevalence of

depression and depressive

symptoms among outpatients:

a systematic review and

meta-analysis. BMJ Open

2017;7:e017173. doi:10.1136/

bmjopen-2017-017173 for depression diagnosis in TN patients

3. https://doi.org/10.1016/j.genhosppsych.2021.01.006

4. https://doi.org/10.1038/s41392-024-01738-y

We have re-worked the introduction extensively (Lines 104-126) in an attempt to better capture the complexity of the underlying mechanisms of the TB-depression syndemic, and have now included the example of co-infection with HIV as an example of a potential underlying mechanism. We greatly appreciate your suggestion of references as well as those of Reviewer #1 and hope that our re-working of the introduction has now addressed your concerns.

I am unable to grasp the message conveyed by this statement.

We apologize for any lack of clarity. We have re-worked the sentence in question to the following (lines 130-131):

From a public health perspective, untreated depression contributes to delays in TB diagnosis and treatment initiation (26, 28), non-adherence to TB medications (29, 30), and ultimately increases negative TB outcomes such as loss to follow up and death (31, 32).

This last part of the introduction should be revised to clearly state the rationale for conducting a scoping review on this topic and the aim of this protocol article. Based on the cover letter and the fact that the authors will use the RE-AIM framework for data synthesis, the rationale for this scoping review seems to be the need to guide future studies by providing a roadmap for integrating appropriate (from a public health perspective) diagnostic and therapeutic interventions for depression into routine care of different forms of TB: see Munn et al. BMC Medical Research Methodology (2018) 18:143

https://doi.org/10.1186/s12874-018-0611-x.

Alonmg with my comment about the rationale for the scoping review, a simpler delineation of the aim of this scoping review protocol article could be: to evaluate the public health impact of diagnostic and therapeutic interventions for depression in individuals with different forms of TB: see PMID: 10474547 PMCID: PMC1508772 DOI: 10.2105/ajph.89.9.1322.

Thank you for this suggestion. We have updated the final paragraph of the introduction as follows (lines 165-175):

Given the substantial burden of depression among PWTB and the large focus on prevalence estimates within the literature with limited exploration of the methodological diversity and public health relevance of diagnostic and therapeutic approaches, a scoping review is particularly needed to broadly map the existing evidence, identify the most pressing knowledge gaps, and clarify how research has been conducted when studying these comorbid conditions. A scoping review would also help guide future studies by offering a roadmap for integrating effective and contextually appropriate mental health interventions into TB care. Therefore, we aim to conduct a scoping review to examine the methodologies used to study depression, including diagnostic and treatment methodologies, in PWTB, identifying research gaps and highlighting opportunities for improved methodological approaches in future public health research.

Because this is the protocol (not the scoping review) article, I suggest revising this statement in these likes:

"This section is presented in accordance with the scoping review framework by Arksey and O'Malley [26] and refined by Levac et al [27] which will guide the methodology of our scoping review."

Did you also register the protocol in OSF for example? Pieper and Rombey Systematic Reviews (2022) 11:8

https://doi.org/10.1186/s13643-021-01877-1

Thank you. We have revised the sentence per your suggestion. We have not registered the protocol in OSF yet but as is now mentioned in manuscript (lines 326-329) we will be making all relevant data from the review available in OSF upon completion of the review and will have the protocol registered prior to data extraction

Along with the potential rationale and aim of the scoping review mentioned above, is this question not more exact:

"What methods have been used so far to diagnose and treat depression across different forms of TB and did those methods have sizeable public health effects?

We thank you for your consideration of our research question. We believe that our current research question captures the aim of our review as we will not be able to answer the second part of your proposed question (“did those methods have sizeable public health effects”) since we will not be conducting a meta-analysis of any diagnostic tools or treatment interventions. We are seeking to have a broad inclusion to see how researchers have tried to define the relationship between TB and depression. Studies may also be characterizing the depression risk profile of those with TB which would also be of interest beyond just treatment and diagnosis. That being said, depending on the results of the present scoping review, conducting a follow-up systematic review and meta-analysis looking at the public health impact of treatment interventions may be an appropriate next step.

Consider deleting this statement to leave only the research question in this sub-section.

We have deleted the sentence “We will screen studies that have sought to examine the relationship between TB and depression”

Can you reframe this statement to first enumerate the searched electronic databases and then state the reasons why you are planning to search those databases?

We have replaced the noted sentence with the following (lines 194-202):

We will search the following electronic databases: MEDLINE, Embase, Global Health, the Cochrane Library, the WHO Regional Libraries, Africa-Wide Information, and PsycINFO. These databases were selected for their relevance to the scope of this review: MEDLINE, Embase, Global Health, and the Cochrane Library are widely used for biomedical and public health research; the WHO Regional Libraries and Africa-Wide Information are included to ensure geographic representation, particularly from high TB burden regions such as sub-Saharan Africa; and PsycINFO is included for its comprehensive coverage of psychological and mental health research.

Could you revise this statement to clearly specify that the electronic search will be supplemented by manual searches consisting of the scrutinization of reference lists of studies retrieved from electronic search? Will you also conduc grey literature searches? Notably, I anticipate that there are non-peer reviewed/non-published materials addressing this topic in a subset of African TB/mental health settings.

We have updated the referred to sentence to the following (lines 219-222):

Finally, the electronic database search will be supplemented by manual searches, including the screening of reference lists from all included studies to identify any additional relevant articles that may have been missed.

We appreciate the suggestion to include grey literature and strongly considered it prior to the initial submission as well as again now. However, given the methodological focus of this scoping review and the resource constraints of our research team, we have chosen to limit our search to peer-reviewed literature to ensure consistency in reporting standards and methodological detail, which are often less clearly documented in grey literature sources, as we need studies fully detailing their methods in order to answer our research question.

Consider adding details about participants' age, sex and socioeconomic status (JBI Evid Synth. 2020 Oct;18(10):2119-2126. doi: 10.11124/JBIES-20-00167.)

Consider also mentioning a statement about comparators of participants with the TB-depression syndemic given that you will include interventional studies.

Within Table 1, we have added in wording to explicitly state that participants of any age, sex, or SES will be included.

I suggest reframing this statement like this:

"human participantss with only a clinical diagnosis of tuberculosis, those with a mycobaterial infection not due to M. tuberculosis (e.g., leprosy), and animal participants"

The fact that the authors also include participants with latent tuberculosis mean that they cannot e

---

## [Decision Letter · Decision Letter 1]

PONE-D-25-16622R1Methodologies for studying depression in persons living with tuberculosis: Protocol for a scoping reviewPLOS ONE

Dear Dr. Gupta,

Thank you for submitting your manuscript to PLOS ONE. After careful consideration, we feel that it has merit but does not fully meet PLOS ONE’s publication criteria as it currently stands. Therefore, we invite you to submit a revised version of the manuscript that addresses the points raised during the review process.

We look forward to receiving your revised manuscript.

Kind regards,

Mickael Essouma, M. D.

Academic Editor

PLOS ONE

Journal Requirements:

Additional Editor Comments:

The authors have improved the manuscript and the reviewer has recommended it to be accepted for publication. However, careful editing is required before an accept decision can be issued.

More details are provided in the document PONE-D-25-16622_R1_Mickael Essouma comments attached to this decision letter.

Reviewers' comments:

Reviewer's Responses to Questions

**Comments to the Author**

1. Does the manuscript provide a valid rationale for the proposed study, with clearly identified and justified research questions?

Reviewer #1: Yes

2. Is the protocol technically sound and planned in a manner that will lead to a meaningful outcome and allow testing the stated hypotheses?

Reviewer #1: Yes

3. Is the methodology feasible and described in sufficient detail to allow the work to be replicable?

Reviewer #1: Yes

4. Have the authors described where all data underlying the findings will be made available when the study is complete?

Reviewer #1: Yes

5. Is the manuscript presented in an intelligible fashion and written in standard English?

Reviewer #1: Yes

6. Review Comments to the Author

You may also provide optional suggestions and comments to authors that they might find helpful in planning their study.

Reviewer #1: Dear authors,

Thank you for thoroughly addressing all of my comments. I believe the paper is ready for publication in its current form. I have just one minor suggestion: please ensure that all abbreviations are properly introduced—such as "RE-AIM" in the abstract.

You've done an excellent job with this work, and it was a pleasure to review the protocol. Thank you for the opportunity.

Best regards,

Max Berg

7. PLOS authors have the option to publish the peer review history of their article (what does this mean? ). If published, this will include your full peer review and any attached files.

**Do you want your identity to be public for this peer review?** For information about this choice, including consent withdrawal, please see our Privacy Policy .

Reviewer #1: **Yes: ** Max Berg

---

## [Author Response · Author response to Decision Letter 2]

17 Jun 2025

We would like to once again thank the editor and reviewer for careful consideration of our manuscript. Many thanks to Dr. Max Berg for his approval of the current version. We believe we have now addressed the concerns of Dr. Essouma as well. We have outlined changes and responses below.

To Dr. Essouma:

Thank you for your detailed feedback. For ease of incorporating revisions and ensuring we don’t overlook any of your comments, would you mind providing any additional feedback after this version using a Word document and citing line numbers rather than a PDF? We greatly appreciate your time and suggestions, and working in Word will help us respond more efficiently and accurately.

“You tend to make long sentences. Think about reversing this tendency by rather increasing the number of simple sentences in the manuscript. Consider also being consistent with the use of abbreviations, not to mention the need for full spelling of abbreviations when first using them.

'etc' is not warranted in the manuscript."

We have gone through and done as you suggest by shortening sentences and removed the use of ‘etc’.

Also to note, there were many spots where you have suggested certain wording. We have updated per your wording suggestions in the abstract and throughout the manuscript.

Where are the keywords?

In the documentation you had previously suppled about formatting per PLoS ONE guidelines, the keywords were not included within the manuscript itself but simply in the manuscript data fields. In this case the keywords are: tuberculosis; depression; global health; reviews. They can be found on the first page of the pdf in the six row of the table.

This introduction needs to be shortened by eliminating redundant statements for example. This will help put more emphasis on the methods section which is the most important section of a protocol article such as this one. Even the reference count of the introduction would better be reduced, notably the number of references addressing the relationship between depression and TB. Consider keeping only the most recent highest-quality data synthesis articles or primary study articles where data synthesis articles are not available.

We had previously expanded the introduction per the request of the reviewers. This included greatly expanding the reference count and included adding in references specifically requested by the reviewers. Given that this was an explicit ask of the reviewers, we would like to leave most of the added references in to maintain the approval of the reviewers. However, we have now shortened the introduction as you suggest and believe this puts more emphasis on the methods section.

Consider supporting the statements on latent tuberculosis with at least one reference.

We have now added in a reference at the designated location (Line 64).

I still make this proposition to remove some statements below in order to reduce the length of the introduction.

We have now shortened the introduction while trying to maintain the requested edits by the reviewers. We hope we have now struck a better balance between giving complete information while being succinct.

This statement is implicitly repeated in lines 90-92. This is the reason why I propose this modification in lines 81-83.

We have made the change as you proposed although have left the relevant references as they are both recent and relevant.

Given that the sentence in lines 97-99 contains a bit of information provided in lines 77 and 78 and widespread knowledge that HIV infection naturally leads to immunosuppression, I suggest abbreviating this statement like this:

"TB-depression co-occurrence can be thought of as a syndemic, or an interacting set of both conditions that cluster within a constellation of biological (e.g., HIV infection) and social factors (e.g., stigma, economic vulnerability) factors capable of heightening susceptibility to mental health disorders."

I did not use the word "structural" because I thought "structural factors" may be embedded within "social factors".

We have updated this sentence to the following (Lines 99-102) which we believe captures your intent while maintaining the emphasis on the definition of a syndemic:

TB-depression co-occurrence can be thought of as a syndemic, or an interacting set of conditions that cluster within social disadvantage (e.g. stigmatization of health conditions and economic vulnerability) that results in an increased burden of disease (23-25).

Consider merging this paragraph with the previous one because the idea developed in this paragraph and that developed in the previous one are in the same vein and this will help decrease the length of the introduction.

After much consideration, we have decided to maintain the current paragraph structure. This avoids having an extremely large paragraph and will improve readability. We thank you for the suggestion and as mentioned previously, we have worked to reduce the size of the introduction.

I propose the deletion of this paragraph because this information has been provided in lines 70-74 though not using the exact same words.

The paragraph in question specifies various mechanisms of the interaction between depression and tuberculosis. The lines that you reference here (lines 70-74) are specific to TB and do not touch on the interplay between the two diseases. Given this, we believe the readers will be expecting a simple synthesis of the known interactions between depression and tuberculosis so have chosen to leave this paragraph in. We have made of wording changes to further reduce the length of the introduction but believe this is an important section for synthesizing what has been previously stated.

This information has already been provided above and is also provided in lines 284-287 though not using the exact same words. I moved a section of this text "appears to have been limited to prevalence surveys" to line 90. Therefore, consider deleting this statement here.

Unfortunately, we are not entirely sure which lines you are referencing as 284-287 is in the discussion and lines 84-87 do not provide comparable information. We have however altered this paragraph to be more concise (lines 179-184):

Despite the significant impact of depression in PWTB, research has largely been limited to prevalence surveys. Few studies have used longitudinal designs and even fewer explore the methodologies used to screen, diagnose, or treat depression in PWTB. This may be in part due to diagnostic challenges posed by overlapping symptoms, such as TB-related fatigue, weight loss, and general malaise, all of which can masquerade as signs of depression, leading to misdiagnoses or missed diagnoses.

I suggest also deleting this statement (lines 118-131) from the introduction because some bits of information provided here have already been provided above though not using the exact same words. Furthermore, comments about diagnostic challenges and the need for complex systems thinking and multiple perspectives in the diagnosis and management of depression in PWTB would better be moved to the discussion section.

We originally added this into the introduction at the request of a reviewer. We have now however eliminated this discussion from the introduction.

The verb tenses in the methods, discussion and conclusion sections should be consistent with the information from the timeline sub-section.

We believe we have now updated tense use appropriately based on the timeline.

Because based on the information in line 269, it seems that you have started searching the literature for the scoping review.

You are correct. We have placed the search in the past tense now.

MeSH are only for PubMed, not the other databases. I therefore suggest deleting this sentence and referring readers to S1 appendix for the full search strategies used in different electronic databases at the end of the previous sentence.

Thank you for this comment. While it is correct that MeSH terms are specific to PubMed, we note that several other databases included in our search strategy—such as Cochrane Library and PsycINFO—use controlled vocabularies that are either derived from or aligned with MeSH terminology. However, to avoid any potential confusion, we have removed the sentence in question and now refer readers directly to S1 Appendix for the complete search strategies used across databases. We appreciate the opportunity to clarify this point.

Which laboratory samples? Specify, remembering that you said you will include even only microbiologically confirmed TB.

In this case, the exclusion criteria means we will not be included “bench” studies. So for example studies that are utilizing blood or sputum samples to test new diagnostics or therapeutics but do not involve a clinical component. Many of these studies are focused on immune interactions and cellular responses which are not the focus of the current scoping review. This is why we have specified studies that are only looking at laboratory samples and isolates. Any individual with TB diagnosed in any way will be included with most PWTB being diagnosed microbiologically.

I would replace this term with "Main variables"

This is about the word “Outcome” used in our Table 1. However, we would like leave it as “Outcome” as this is the standard terminology utilized for reviews (Population, Intervention, Comparator, Outcome, Timing, Setting; PICOTS table).

Revise this sub-section with simple sentences. Accordingly, you could summarize the data to be collected in a table inserted in the main manuscript (Table 2). This would shorten this sub-section while making it more readable.

We have done as you advise and switched the majority of the “Charting the Data” paragraph into a Table, now labeled Table 2: Data Extraction.

You would better present overall data and segregated data (segregation by age category [infants, children, adolescents, adults, and older adults] sex, gender, race, ethnicity, geographic location [continents/world regions] to name a few) and you need to mention that information in this sub-section as well.

We have added in the following (Lines 358-360):

We will provide descriptive characteristics of all included studies using figures and tables to fully summarize the studies, stratifying where appropriate by key variables such as by age, TB type, geographic location and/or study design to name a few.

Given that this paragraph is very long and difficult to follow and considering the importance of well explaining how the RE-AIM approach will be used in your scoping review, I suggest explaining how you will use the RE-AIM approach in the scoping review using a figure or a table inserted in the main manuscript. Alternatively, consider subdividing this paragraph into two paragraph including simple sentences.

We have now shortened this paragraph and split it into two paragraphs.

Consider revising this statement with the discussion section to avoid repeating information available here in the discussion section

We have now shortened the referred to sentences (lines 416-418) to avoid replication of later sentences:

Consultations will offer contextual insights and help identify gaps or implementation issues not captured in the published literature, ensuring the findings are meaningful and actionable for high TB burden settings.

There is also a need to shorten the discussion section by removing redundancy. The revisions proposed in the discussion therefore aim to abbreviate its text while keeping its main content unchanged.

We had similarly expanded the Discussion to address concerns of the reviewers. We appreciate your assistance in identifying areas where we can be more concise and believe we have also further cut down on its size.

I suggest deleting this statement because it repeats the information provided in lines 278 and 279, and the information "provide insights into potential pathways for improving integrated care" is somewhat provided in lines 290 and 291 using different words.

We have eliminated as you suggest.

Given that this statement discusses an idea that is quite different from the idea discussed in the preceding sentence, I suggest you go to the line from here.

It seems to me that the authors are discussing the strengths of the upcoming scoping review here. Therefore, I suggest you go to the line here because this idea is quite different from the one discussed in the previous sentence. Furthermore, consider revising the beginning of this statement in these likes:

"Strengths of our review will include a methodological rigour based on

We have started a new paragraph at the requested spot (Line 467) and adjusted the sentence as you requested.

This paragraph would better be shortened.

Thank you for this suggestion. This comment refers to the limitations paragraph. While we have implemented some of the suggested edits to improve clarity, we have retained much of the original content, as this section was specifically highlighted by both peer reviewers as a strength of the manuscript. We believe preserving this version maintains alignment with their positive feedback.

The terms "diagnosis" and "treatment" appear nowhere in the conclusion?

I also notice that from the introduction to the conclusion, you focus on the fact that the scoping review will help you find research gaps, but will it not first help you find the "state of knowledge" on methodologies used to assess depression diagnosis and treatment before helping you find the gaps in knowledge? Consider addressing this issue throughout the manuscript.

Thank you for this thoughtful observation. We agree that the primary aim of the scoping review is to first map the current state of knowledge regarding methodologies used to study depression diagnosis and treatment in individuals with TB, and then to identify key gaps. While this intent is reflected throughout the manuscript, we acknowledge that the conclusion could more explicitly echo this two-step purpose. We have revised the conclusion slightly to incorporate the terms "diagnosis" and "treatment" and to more clearly emphasize that the review will both summarize existing methodological approaches and highlight areas where further research is needed (Lines 540-543). We believe this clarification strengthens alignment across the manuscript.

This scoping review will advance understanding of how depression screening, diagnosis, and treatment have been studied in the context of tuberculosis by systematically mapping existing methodologies and identifying critical research gaps to guide future public health research and intervention development

---

## [Editor Report · Decision Letter 2]

Methodologies for studying depression in persons living with tuberculosis: Protocol for a scoping review

PONE-D-25-16622R2

Dear Dr. Gupta,

We’re pleased to inform you that your manuscript has been judged scientifically suitable for publication and will be formally accepted for publication once it meets all outstanding technical requirements.

Kind regards,

Mickael Essouma, M. D.

Academic Editor

PLOS ONE
---

## [Editor Report · Acceptance letter]

PONE-D-25-16622R2

PLOS ONE

Dear Dr. Gupta,

I'm pleased to inform you that your manuscript has been deemed suitable for publication in PLOS ONE. Congratulations! Your manuscript is now being handed over to our production team.

Kind regards,

on behalf of

Dr. Mickael Essouma

Academic Editor

PLOS ONE